# Rubble Mines in the Environs of Veszprém (Bakony Region, Hungary)

Márton Veress

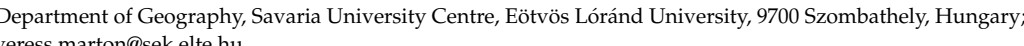

Department of Geography, Savaria University Centre, Eötvös Lóránd University, 9700 Szombathely, Hungary; veress.marton@sek.elte.hu

**Abstract:** In the Bakony Region, in the mines of dolomite (dolostone) surfaces between the settlements of Márkó and Pétfürdő (Várpalota), in rubble beds exposed by them and with the consideration of these, the process of rubble formation is studied here in order to interpret the characteristics of rubble beds (different thicknesses and vertical changes in grain size) in the studied area. The mines in the area (differentiated between old-school/traditional mining or mechanical mining) were classified with the consideration of mining methods. Rubble varieties were differentiated, the bedding of rubble beds was studied along profiles, and the elevation difference between mines of mechanical mining and Stream Séd was determined. The calcareous content and structure compactness of 124 samples originating from dolomite, rubble, and non-rubble in the Bakony Region were compared. The data prove that the rubble developed by dissolution. Dissolution might have been caused by both meteoric water and karstwater. The rubble of mines excavated by traditional mining mainly developed to the effect of the dissolution effect of meteoric water (the rubble beds are of coarser and coarser grain size downwards), while the mines excavated by mechanical mining were formed to the dissolution effect of karstwater (the rubble beds are coarser and coarser upwards). The formation of rubble by karstwater origin has not been mentioned in the literature yet. However, dissolution of meteoric water origin may also take place in the case of the latter, and dissolution of karstwater origin also plays a role in the development of mines excavated by traditional mining.

**Keywords:** dolomite; rubble; mine; dissolution by meteoric water; dissolution by karstwater

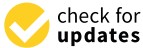



## 1. Introduction

In this study, the rubble formation of dolomite is studied in order to interpret the characteristics (different thicknesses and vertically different changes in grain size) of rubble excavated by mines. During its denudation, the rubble is separated into pieces and undergoes rubble formation. Current research does not differentiate between the landscapes of dolomite and limestone karsts. Below, the differences between the landscapes of some dolomite terrains and limestone karsts will be discussed. The key to this difference is provided by the excavated rubble beds of the mines of the studied area.

With the study of the rubble of mine walls, data on the trends of the changes in rubble grain size and rubble structure are obtained, but the lack of epikarst can also be established in the studied area. Based on the data, the role of dissolution is described in rubble formation as well as its types (dissolution by surface waters and dissolution by karstwater), the relationship between dissolution and the lack of epikarst, and the relationship between karstwater level and grain size of rubble is also described.

The results of the study classify the stone mines around Veszprém, describe the rubble beds of mine walls, taking into consideration grain size, structure, and position, and describe the relationship between the mine type and the characteristics of the rubble. As compared to earlier reasons triggering rubble formation, new ones are described (fractures) and used (crystal structure). Taking into consideration the characteristics of the mines,

dissolution types resulting in rubble formation are described, but dissolution of karstwater origin has not been mentioned in the literature so far.

Among sedimentary rocks, sedimentary rocks with debris are notable. Within these rocks, those with a predominant grain size over 2 mm are called loose, rag sediments with debris [1]. The material of the dolomite that is separated into grains and the rubble belongs to this interval; however, it cannot be regarded as a sedimentary rock (at most it is weathering residue that developed in situ) because it did not develop during transportation, during condensation from solution, or during the accumulation of organic matter. However, in colloquial language, in mining and applied geological research, an appearance of dolomite is distinguished that developed in situ, separated into parts, but with changing grain size and it is called rubble, powdered rubble, and dolomite of non-uniform development [2]. Below, that cover material is regarded as rubble and rubble variety, the grain size of which may range from some millimetres to 1–2 decimetres, which is not rounded, but angulate, and thus, not transported, but developed from solid dolomite in situ.

Mining results in morphological [3,4] and hydrological [4,5] changes but also affects geomorphic evolution [6–8]. Mines may be excavated by underground mining or surface mining. Studies on the relationship between mining and karstic landscapes have become widespread [9–13]. In the Bakony Region, several studies have mentioned the relationship between mining and karst and paleokarst (for example) [14]. Two studies had a complete focus on this relationship in the Bakony Region [13,15].

Mining results in negative (depressions) and positive features (waste dumps) [6,7,16,17]. Positive features are mounds of different shapes and accumulation origins that develop during mining activity. Negative features may develop during surface mining (strip pits) or underground mining. The latter may be formed due to material hiatus [18–21], water extraction [22], and artificial subsidence of water levels [23,24]. Positive features may also develop during opencast mining. Thus, the surface waste dump is bulldozed entrenchment-like features that are formed around the strip pit. However, the waste dump may survive in its original bedding in the area of the strip pit, or it may also be reworked. In the latter case, the mined rubble is piled up into heaps of different sizes and shapes during classification (according to grain size). These heaps are mostly temporary. The morphology of the strip pit is more or less identical to the spatial extension of the mined material.

## 2. Research Area

In this study, the relationship between various rubble mine types and rubble formation is investigated in the area between Márkó and Pétfürdő (Várpalota) (Figure 1).

The research area is situated in the Bakony Region at the boundary of Northern Bakony and Southern Bakony, between the settlements of Márkó és Pétfürdő (Várpalota), which is separated into a western rubble area and an eastern rubble area by the rubble-free part bearing the town of Veszprém. The Bakony Region, which is part of the Transdanubian Mountains, comprises mountains with an altitude of 150–700 m and an extension of 4300 km$^2$, which constitutes a disproportionate synclinal structure [25] between the Little Hungarian Plain and Lake Balaton (Mezőföld), but it has separated into blocks by the present day. The peneplain of the mountains is a Cretaceous tropical karstic peneplain [26]. Its most widespread rock is the Late Triassic main dolomite (Main Dolomite Formation) which developed in the Carnian and Norian stages, but dolomite also developed in the Carnian, Anisian, Ladinian, and Campilian stages. The thickness of the Triassic Main Dolomite exceeds 1500 m at some sites [27]. About 21% of the karst surfaces of the Transdanubian Mountains are constituted by these rocks [28], but its proportion exceeds 24% in the Bakony Region. The composition of dolomite is described in Table 1 based on samples collected by Hegyiné Pakó et al. [2], while its surface distribution in the Bakony Region is shown in Figure 2a,b. Triassic dolomites and limestones are overlain by Jurassic, Cretaceous, and Eocene limestones only in small thicknesses in patchy development. Similarly, the material of the Csatka Gravel Formation is also of patchy development due to denudation by erosion.

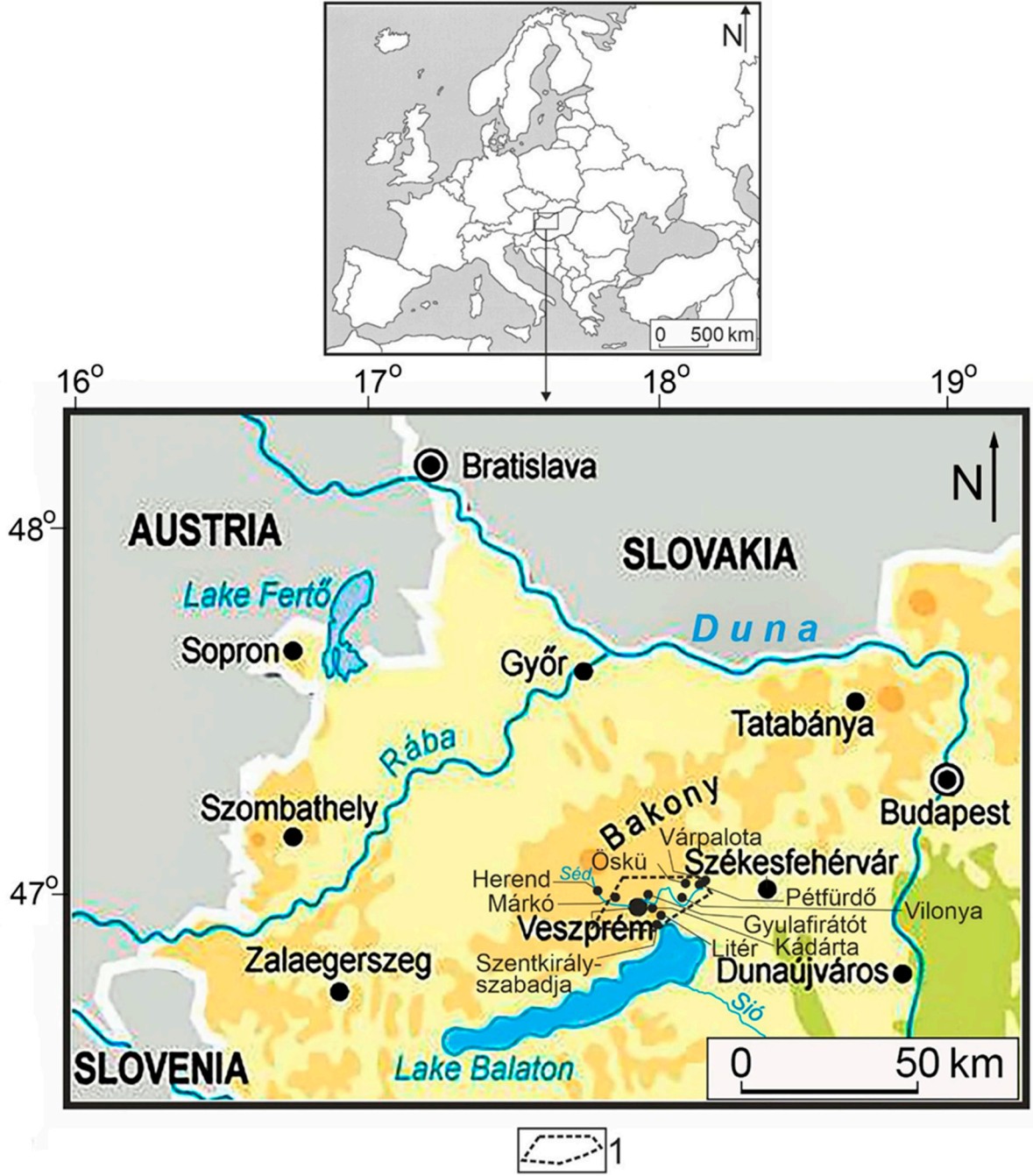

**Figure 1.** Overview map of the studied area. Legend: 1. Research area.

**Table 1.** Chemical characteristic features of rubble beds [2] (modified).

| Sequential Number [1] | Sequential Number [2] | Name | Site | CaO [Mass Fraction %] | Decrease of CaO Mass (%) | MgO [Mass Fraction%] | Decrease of MgO Mass (%) | CaO/MgO Proportion | Mineral Beds [3] |
|---|---|---|---|---|---|---|---|---|---|
| 1 | 62 | main dolomite (Norian) | Markó main road 8 | 32.16 | 67.85 | 20.50 | 79.5 | 1.57 | dolomite (feldspar) |
| 2 | 64 | main dolomite (Norian) | Veszprém mine | 31.80 | 67.2 | 20.99 | 79.01 | 1.52 | dolomite (feldspar) |
| 3 | 68 | calcareous main dolomite (Norian) | Litér mine | 41.87 | 58.13 | 12.8 | 87.2 | 3.47 | dolomite, calcite |
| 4 | 69 [4] | calcareous main dolomite (Norian) | Litér mine | 38.60 | 61.4 | 14.93 | 85.7 | 2.59 | dolomite, calcite |
| 5 | 70 [4] | main dolomite (Norian) | Litér mine | 30.54 | 69.46 | 19.68 | 80.32 | 1.55 | dolomite, calcite (feldspar) |
| 6 | 734 | main dolomite (Norian) | Sóly mine | 32.11 | 67.89 | 20.31 | 79.69 | 1.58 | dolomite (feldspar) |
| 7 | 74 | Megyehegy Anisian | Sóly mine | 32.36 | 67.64 | 19.27 | 80.73 | 1.64 | dolomite (feldspar) |
| 8 | 81 [4] | calcareous main dolomite (Norian) | Várpalota mine | 33.32 | 66.68 | 19.32 | 80.68 | 1.72 | dolomite, calcite (feldspar) |
| 9 | 98 | dolomite (Carnian) | Guttamás mine | 32.14 | 67.86 | 20.46 | 79.54 | 1.57 | dolomite (feldspar) |
| 10 | 99 | dolomite (Carnian) | Guttamás mine | 32.65 | 67.35 | 19.93 | 80.07 | 1.64 | dolomite (calcite, feldspar) |
| 11 | 100 [4] | dolomite (Carnian) | Várpalota mine | 32.27 | 67.73 | 20.06 | 79.94 | 1.61 | dolomite (feldspar) |
| 12 | 101 [4] | dolomite (Carnian) | Hideg valley | 31.76 | 68.24 | 20.69 | 79.31 | 1.54 | dolomite (feldspar) |
| 13 | 102 | Dolomite (Carnian) | Hideg valley | 31.92 | 68.08 | 20.22 | 79.78 | 1.58 | dolomite |

**Table 1.** *Cont.*

| Sequential Number [1] | Sequential Number [2] | Name | Site | CaO [Mass Fraction %] | Decrease of CaO Mass (%) | MgO [Mass Fraction%] | Decrease of MgO Mass (%) | CaO/MgO Proportion | Mineral Beds [3] |
|---|---|---|---|---|---|---|---|---|---|
| 14 | 104 | Megyehegy dolomite (Anisian) | Iszkaszent-györgy mine | 32.61 | 67.39 | 20.09 | 79.91 | 1.62 | dolomite (feldspar) |
| 15 | 105 | main dolomite (Norian) | Öcs mine | 32.10 | 67.9 | 20.50 | 79.5 | 1.57 | dolomite (feldspar) |
| 16 | 108 [4] | main dolomite (Norian) | Öcs mine | 31.91 | 68.09 | 20.54 | 79.46 | 1.55 | dolomite (feldspar) |
| 17 | 111 [4] | main dolomite (Norian) | Zsófia farm | 32.12 | 67.28 | 20.30 | 79.7 | 1.58 | dolomite (feldspar) |
| 18 | 112 [4] | main dolomite (Norian) | Kövesgyőr puszta | 32.14 | 67.86 | 20.56 | 79.44 | 1.56 | dolomite (feldspar) |
| 19 | 116 | main dolomite (Norian) | Balatonalmádi mine | 31.74 | 68.26 | 20.70 | 79.3 | 1.53 | dolomite (feldspar) |

[1] own numbering, [2] numbering according to the above authors, [3] based on thermic roentgen and rock microscope investigations [4] according to the above authors, it was transformed during hydrothermal activity, they used the terms "fractured, rubbled, and powder-like" for the samples [2].

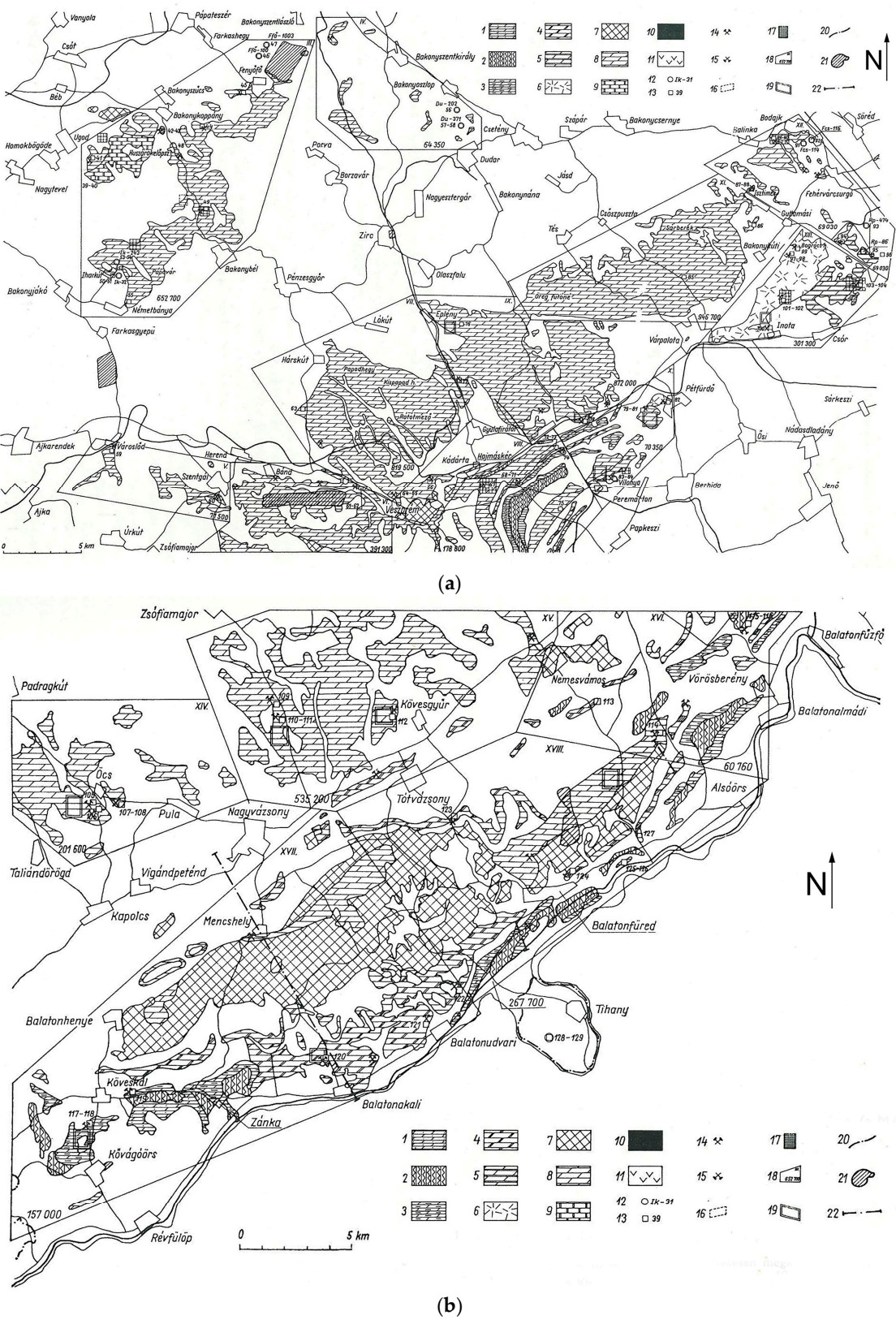

**Figure 2.** (**a**): Distribution of dolomite in Northern Bakony [2]. Legend: 1. Micaceous, sandy (Nádaskút) dolomite with sandstone and schist intercalations (Seizian), 2. Marl, sandstone, dolomite, and limestone, 3. Cellular, porous, thin-bedded, lamellar (Aszófő) dolomite (Campilian), 4. Thick-bedded (Megyehegy) dolomite (Anisian), 5. Diplopora dolomite (Ladinian), 6. Dolomite, 7. Clay marl, limestone, dolomite (Carnian), 8. Layered and bedded main dolomite, calcareous main dolomite,

9. Dolomite limestone, calcareous dolomite, and dolomite intercalations (Carnian-Norian) of hydrothermal origin in limestone, 10. Lamellar, marly, cherty dolomite (Rhaetian), 11. Slope debris with dolomite (Pleistocene), 12. Important drilling that reaches or transverses the dolomite, 13. Site of sampling, sample number, 14. Active dolomite quarry, 15. Abandoned dolomite quarry, 16. Dolomite area under research, 17. Dolomite area excavated in a detailed way, 18. Dolomite area taken into consideration during reserve estimation (reserve of category D per thousand tons), 19. Dolomite area recommended for research in the exploratory phase, 20. Boundary of nature reserve, 21. Strict nature conservation area. (**b**) Dolomite distribution in the Southern Bakony and Balaton Uplands [2]. Legend: 1. Micaceous, sandy (Nádaskút) dolomite with sandstone and schist intercalations (Seizian), 2. Marl, sandstone, dolomite and limestone, 3. Cellular, porous, thin-bedded, lamellar (Aszófő) dolomite (Campilian), 4. Thick-bedded (Megyehegy) dolostone (Anisian), 5. Diplopora dolomite (Ladinian), 6. Dolomite, 7. Clay marl, limestone, dolomite (Carnian), 8. Layered and bedded main dolomite, calcareous main dolomite, 9. Dolomite limestone, calcareous dolomite, and dolomite intercalations (Carnian-Norian) of hydrothermal origin in limestone, 10. Lamellar, marly, cherty dolomite (Rhaetian), 11. Slope debris with dolomite (Pleistocene), 12. Important drilling that reaches or transverses the dolomite, 13. Site of sampling, sample number, 14. Active dolomite quarry, 15. Abandoned dolomite quarry, 16. Dolomite area under research, 17. Dolomite area excavated in a detailed way, 18. Dolomite area taken into consideration during reserve estimation (reserve of category D per thousand tons), 19. Dolomite area recommended for research in exploratory phase, 20. Boundary of nature reserve, 21. Strict nature conservation area.

The extension of the area between Márkó and Pétfürdő (Várpalota) is 10–15 km in the NS direction and 20–25 km in the EW direction. In the studied area, the following dolomites occur at the surface [29]. The Scythian-Anisian Aszófő Dolomite Formation is situated along Litér, Sóly, and Öskü. The Anisian Megyehegy Dolomite Formation is along Litér, Sóly, Öskü, and Hajmáskér. The Ladinian-Carnian Budaörs Dolomite Formation is near Várpalota, but in a tectonic position. The Carnian-Ladinian Main Dolomite Formation, the bedrock of which is the Veszprém marl and the caprock is the Kössen Formation between the Norian-Rhaetian storeys (this is constituted by limestone, dolomite, and marl beds). The main dolomite of the areas of Veszprém, Kádárta, and Szentkirályszabadja constitute a deeper part of the Main Dolomite Formation or it belongs to the Veszprém Marl Formation. In the area, the stratified and bedded calcareous variety of the main dolomite is predominant. Eastwards from Veszprém, the Carnian clay marl, limestone, and dolomite have a smaller and smaller extension (Figure 2a). The main dolomite is stratified only at some sites, it is mostly thick-bedded without impermeable beds.

Structurally, this area is a structural graben with a WE direction which stretches from the settlement of Herend to the eastern margin of the Bakony Region. Its altitude is 160–180 m in Pétfürdő, 190–200 m in Hajmáskér, 190–220 m in Kádárta, and 300–320 m in Márkó. However, in Kádárta, towards the south, the surface altitude exceeds 300 m although there is a decreasing altitude in the eastern direction.

The area is a former pediment [30] or was transformed by pedimentation [31]. Pécsi [32] states that the development of the pediment took place in the Pliocene. The dolomite surface is separated into two levels, an upper and a lower. Terrains belonging to the upper level rise above the terrains belonging to the surrounding lower level. Altitude differences between the two levels are diverse. In some parts of the area, the elevations of the upper level do not necessarily exceed the altitude of a terrain belonging to a lower level which is situated farther away. The two levels primarily differ in their morphologies. The lower level is plain, there is a lack of mounds, and only some smaller valleys are present among indentations. The upper level is dissected by mounds and indentations (these are not valleys) [33]. Rubble beds are widespread in the studied area. In the upper level, hills (mounds) and series of hills of various shapes and sizes as well as elongated indentations reaching 1 km are common, but some closed depressions with small depths also occur [34]. The lower level is covered with some large, plain, irregular surface sections, mainly with

superficial deposits. Mines excavated by mechanical mining are located in these terrains. Terrains dissected by mounds also occur in the environs of Nyirád (Southern Bakony). Triassic dolomite is also overlain by rubble here.

The upper karren part of the karst-containing cavities, the lower part of which is filled with water of fluctuating levels, is the epikarst [35–39]. In the studied area, the epikarst occurs only in some places and locally (Figure 3). The elevation of karstwater is 130–140 m (based on the elevation of springs) in the eastern part of the area (Várpalota), and it is 160 m (based on the isoheights of the karstwater map) to 220 m in the west (Veszprém). The local base level of erosion of the terrain is between Kádárta and Hajmáskér, Stream Séd with WE directions connected by waters of karst springs in the west in Kádárta from the south between the altitudes of 203 and 270 m. Stream Séd and the lakes of the environs, but the springs too refer to the fact that the karstwater level is close to the surface and it is even closer in the east where the elevation of the channel of Stream Séd exceeds the elevation of the karstwater level. Therefore, the latter is the base level of erosion. Stream Séd transports material in eastern and then southern directions towards Mezőföld and Sió, both in suspended and dissolved forms.

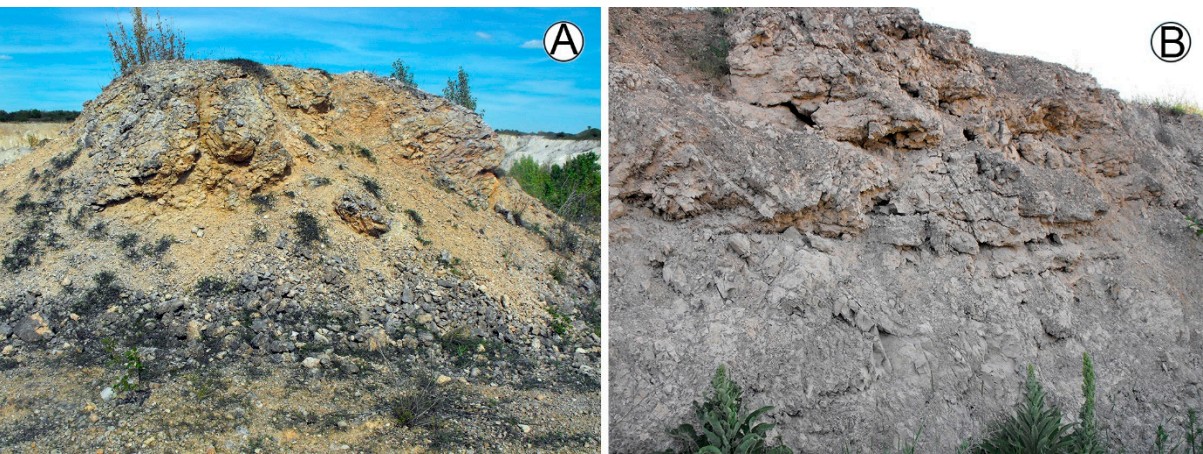

**Figure 3.** Exposed epikarst on a mound of the Sóly strip pit (**A**) and on the mine wall of Kádárta mine (**B**).

### 3. Methods

- Open-cast mines were classified based on their size and morphology. Mines excavated by mechanical mining were described in a detailed way and data on the thickness of rubble were also collected.
- The available samples (data were collected from the work of Hegyiné Pakó et al.) [2] were put into groups. Non-rubble solid dolomite and rubble dolomite samples were differentiated. Samples of both groups were sorted into calcareous (marly) and non-calcareous subgroups and their proportion relative to the total number of samples was also given. Based on grain size, different varieties of dolomite rubble were distinguished.
- The altitudes of mine margins and the closest Séd sections were determined.
- The occurrence of solid rock and rubble relative to each other was also studied in the mines. Based on the profiles of mine walls, bedding varieties were distinguished. Local discolorations of mine walls with various developments were studied in the rubble.
- Fractures of solid dolomite were also studied in rubble and non-rubble places.
- Vertical grain size changes in the rubble of mine walls belonging to various mine types were compared.

## 4. Results

In the studied area, rubble is exposed in road cuts and open-cast mines. Two mine types were distinguished: mines excavated by mechanical mining (Figures 4–6) and traditional mining (Figure 7). Mines excavated by old-school mining (Figure 6) are older (they might have been created in the 19th century or earlier in order to meet the building material demand of local people), they are more common (about 20–30 mines can be found in the studied area), they are smaller and mostly abandoned. Their floor is not plain, they are separated into partial indentations both in profile and on the ground plan, and their margins are arcuate. These mines were made near solid rock exposures, usually in more elevated parts of the dolomite surface; thus, they may be at an altitude of 240–250 m, but at least one mine occurs near the channel of Stream Séd, at the foot of a group of mounds, at an elevation of 200 m. Their morphological environment is also diverse: they occur on the floor of elongated indentations, on more extended mounds (hills), at the margin of mounds reaching the valley floor of Stream Séd, and on the sides of valleys reaching Stream Séd. In the mines, rubble is exposed in a thickness of less than 10 m (sometimes its thickness is only several meters). No caprock occurs in their environment (only soil occurs). In strip pits and mine walls, there are solid rock outcrops that hinder rubble mining (Figure 8). On the upper part of the walls, rubble of medium grain size may be mixed with fine-grained rubble. These mines were selected in sites without superficial deposits due to economic considerations.

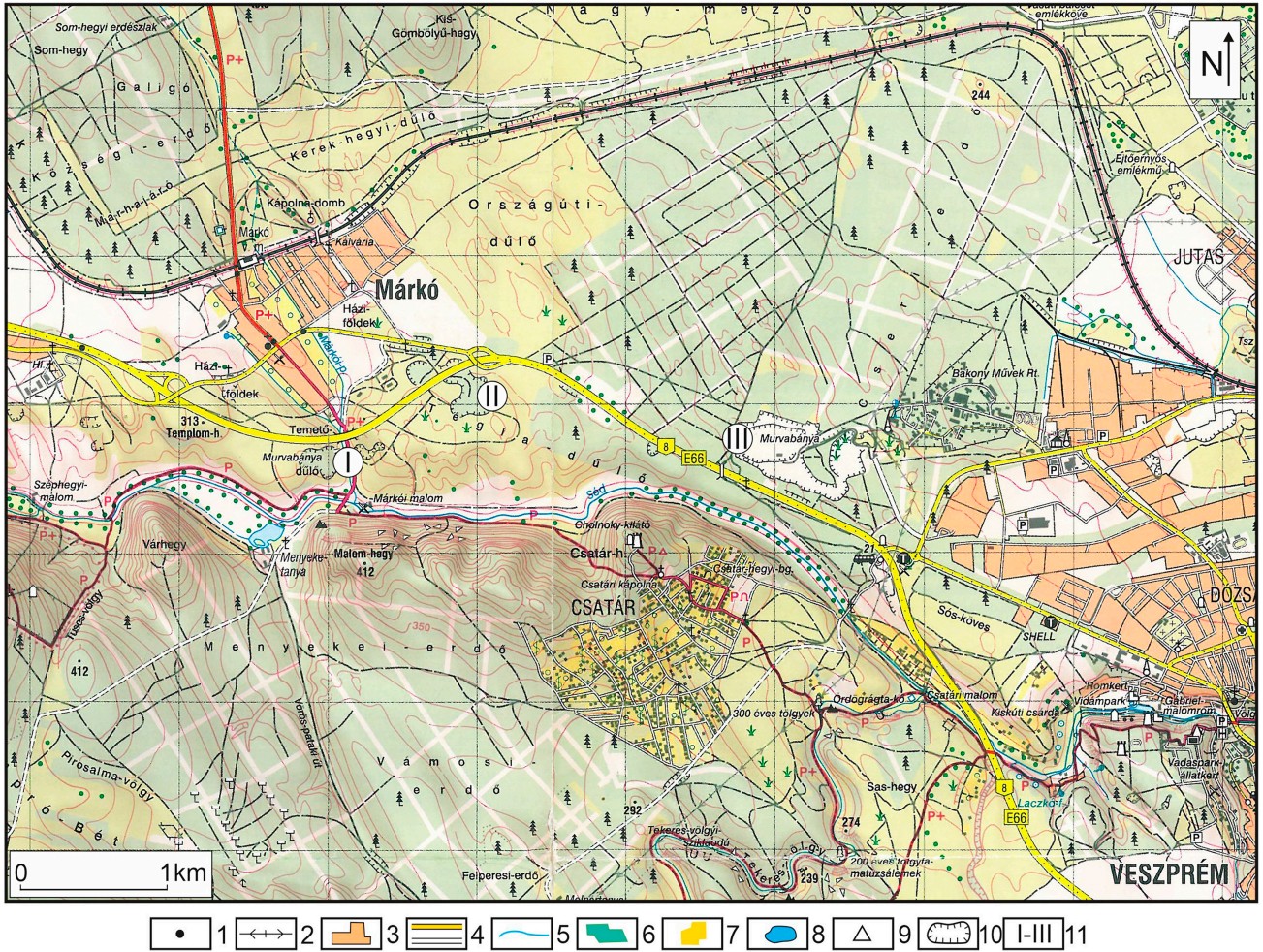

**Figure 4.** Mine occurrences westwards from Veszprém (based on the map 1:40,000). Legend: 1 mountain, 2. power-circuit, 3. settlement, 4. road, 5. stream, 6. forest, 7. grassland, 8. lake, 9. benchmark, 10. mine, 11. mine excavated by mechanical mining (I. Márkó1, II. Márkó2, III. Veszprém mine).

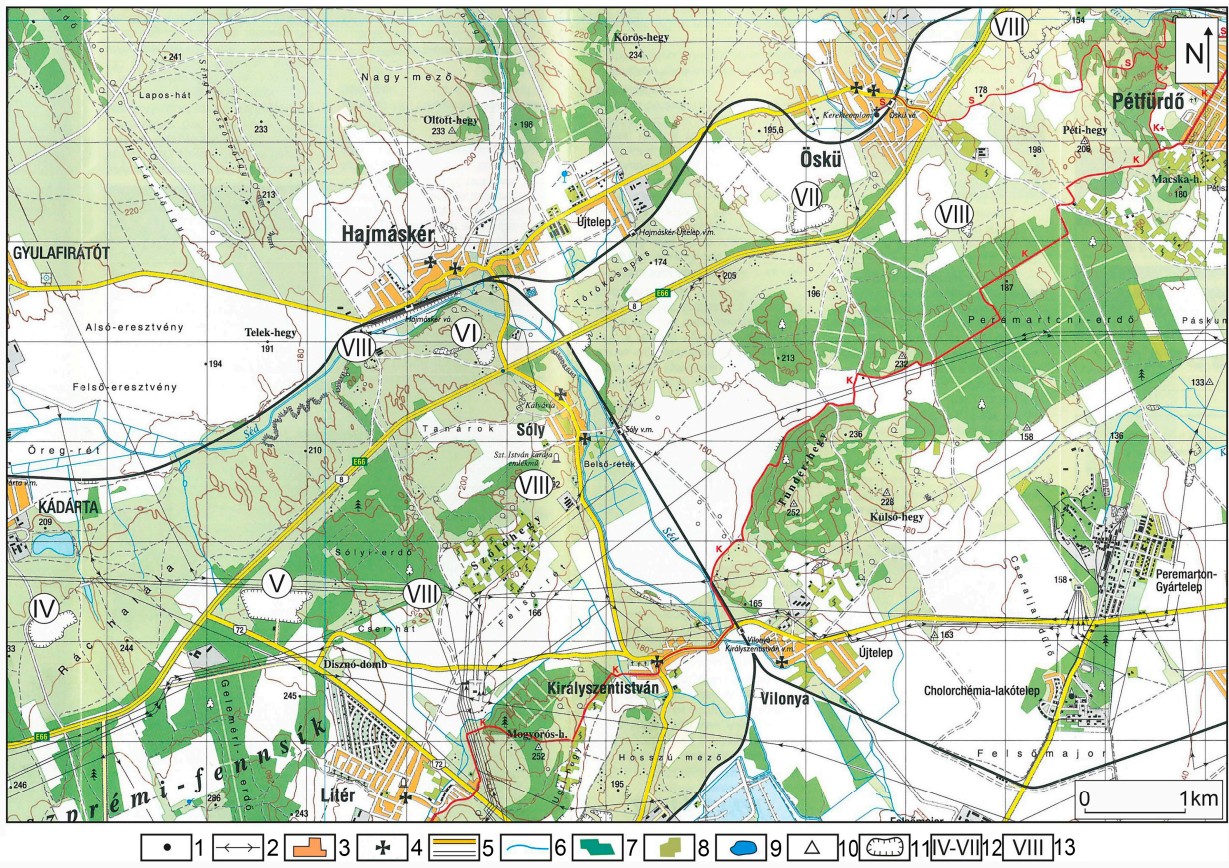

**Figure 5.** Mine occurrences eastwards from Veszprém (based on the map 1:40,000). Legend: 1 mountain, 2. power-circuit, 3. settlement, 4. church, 5. road, 6. stream, 7. forest, 8. grassland, 9. lake, 10. benchmark, 11. mine, 12. mine excavated by mechanical mining (IV. Kádárta mine, V. Litér mine, VI. Hajmáskér mine, VII. Sóly mine), 13. mine excavated by traditional mining.

Mines excavated by mechanical mining are in valley sides westwards from Veszprém, while eastwards from Veszprém, they occur on the large-extended terrains with poor runoff or on the areic, plain, hardly dissected terrains of the lower level. The caprock with a thickness of several meters is highly visible at the margin of mines excavated by mechanical mining (Figure 6). Due to mechanical mining, the presence or lack of caprock on the rubble was probably not a significant factor when the sites of the mines were chosen. The former caprock of operating mines was bulldozed. The so-developed dump entrenchments can be recognized in the environs of the mines. The cover is reworked clay or a mixture of clay, gravel, and loess. The majority of mechanical mines are operating today. The beginning of their development is before the 1970s (except the Márkó 1 mine as topographic maps older than the 1970s do not describe them because they were not created until that time). The mines were made in order to meet the building material demand of the surrounding and farther settlements. These mines are large and are dissected by larger and smaller heaps of extracted rubble, their spatial shape is a geometrical body (cuboid or truncated pyramid), their floors are plain (however, the elevation of floor parts can be different due to mining in several levels), their margins are straight, and tributary mines with arched deepening rarely occur on them. The mines exploit rubble with a thickness larger than 10 m (Table 2). Great rubble thickness is due to not only mechanical mining but also to the fact that the solid rock, which is exposed at some sites in the strip pit, enabled the excavation of rubble at greater depths (for example at the western wall of the mine marked Márkó 1, Figure 9). The grain size of the rubble changes both horizontally and vertically on the mine walls (Figure 10). Commonly, upwards, the rubble does not become fine-grained, but it is coarser and coarser. Thus, in some parts of the mine marked Márkó1, coarse-grained rubble is situated at the topmost place, or in the case of the

Kádárta mine, solid dolomite is at the top. In the studied area, the mines that are excavated by mechanical mining constitute a row with an EW direction (Figures 4 and 5) along Stream Séd. Some mines are described below.

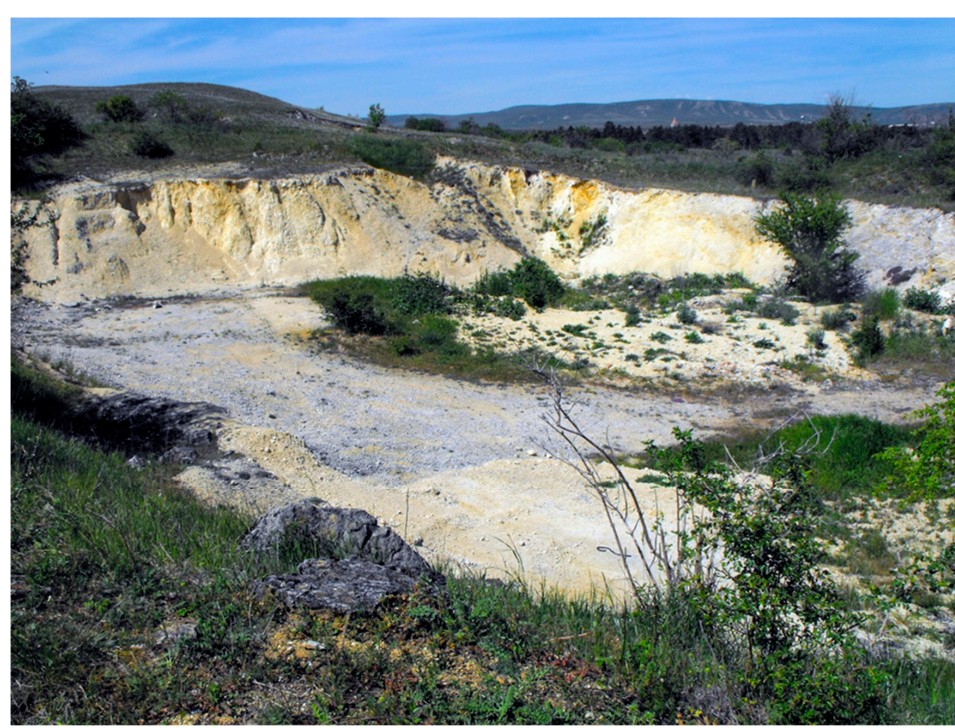

**Figure 6.** Mine excavated by traditional mining.

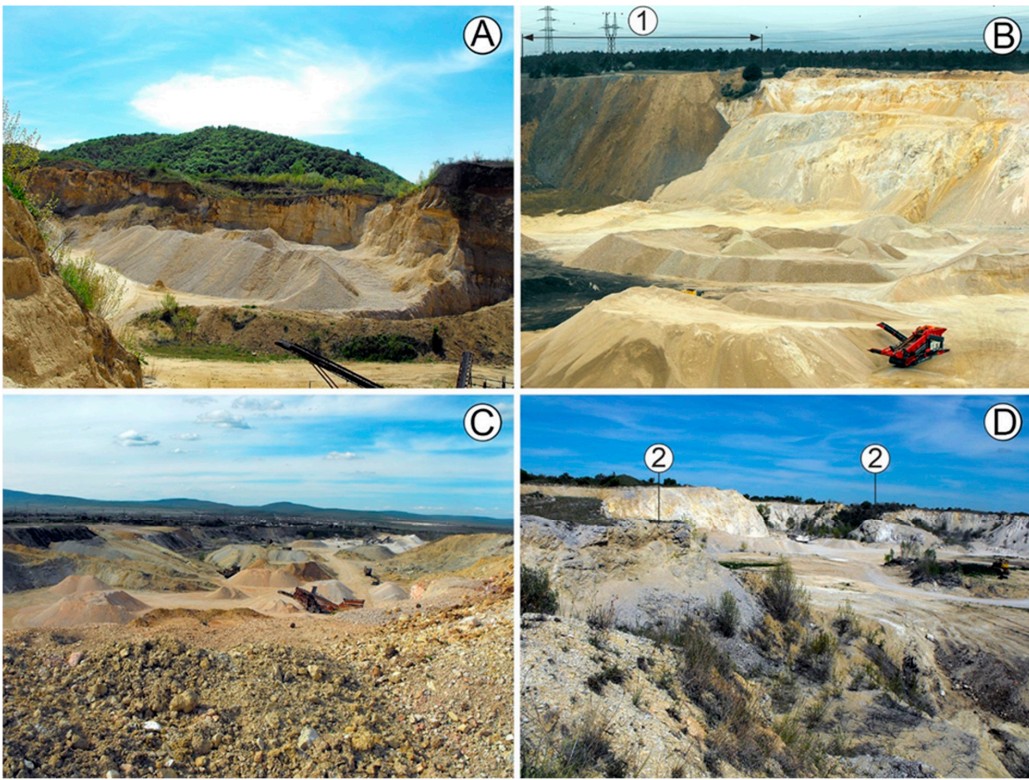

**Figure 7.** Mine excavated by mechanical mining (**A**) Márkó mine, (**B**) Litér mine (**C**) Kádárta mine (**D**) Sóly mine. Legend: 1. bauxite fill of the depression of the dolomite, 2. dolomite mound on the strip pit (solid rock is dirt for rubble).

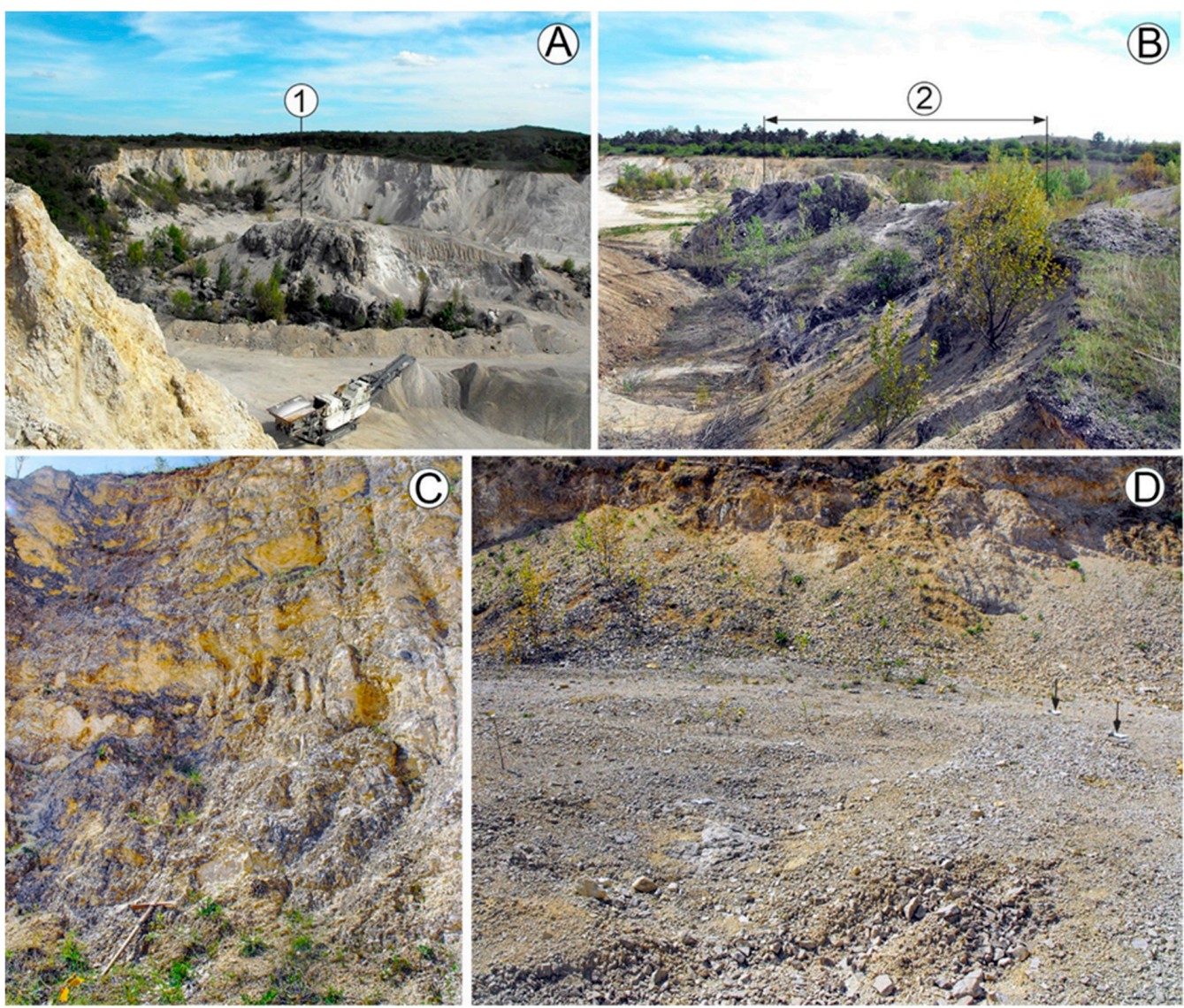

**Figure 8.** Solid rock outcrops (**A**) mound, Sóly mine, (**B**) mound Sóly mine, Legend: 1. solitary dolomite mound, 2. dolomite mound which is connected to the mine margin, (**C**) northern wall of Márkó mine (**D**) floor of the strip pit of Márkó mine (arrows mark the dolomite that crops out from below the rubble, where pieces of paper justify the outcrop).

**Table 2.** Rubble thickness in mines excavated by mechanical mining (mining data reporting).

| Mine | Rubble Thickness [m] | Notice |
| --- | --- | --- |
| Markó1 | 20–25 | Clayey |
| Markó2 | 10 | Clayey |
| Veszprém | 15 | |
| Kádárta | 20 | |
| Litér | 30–35 | |
| Hajmáskér | 20 | |

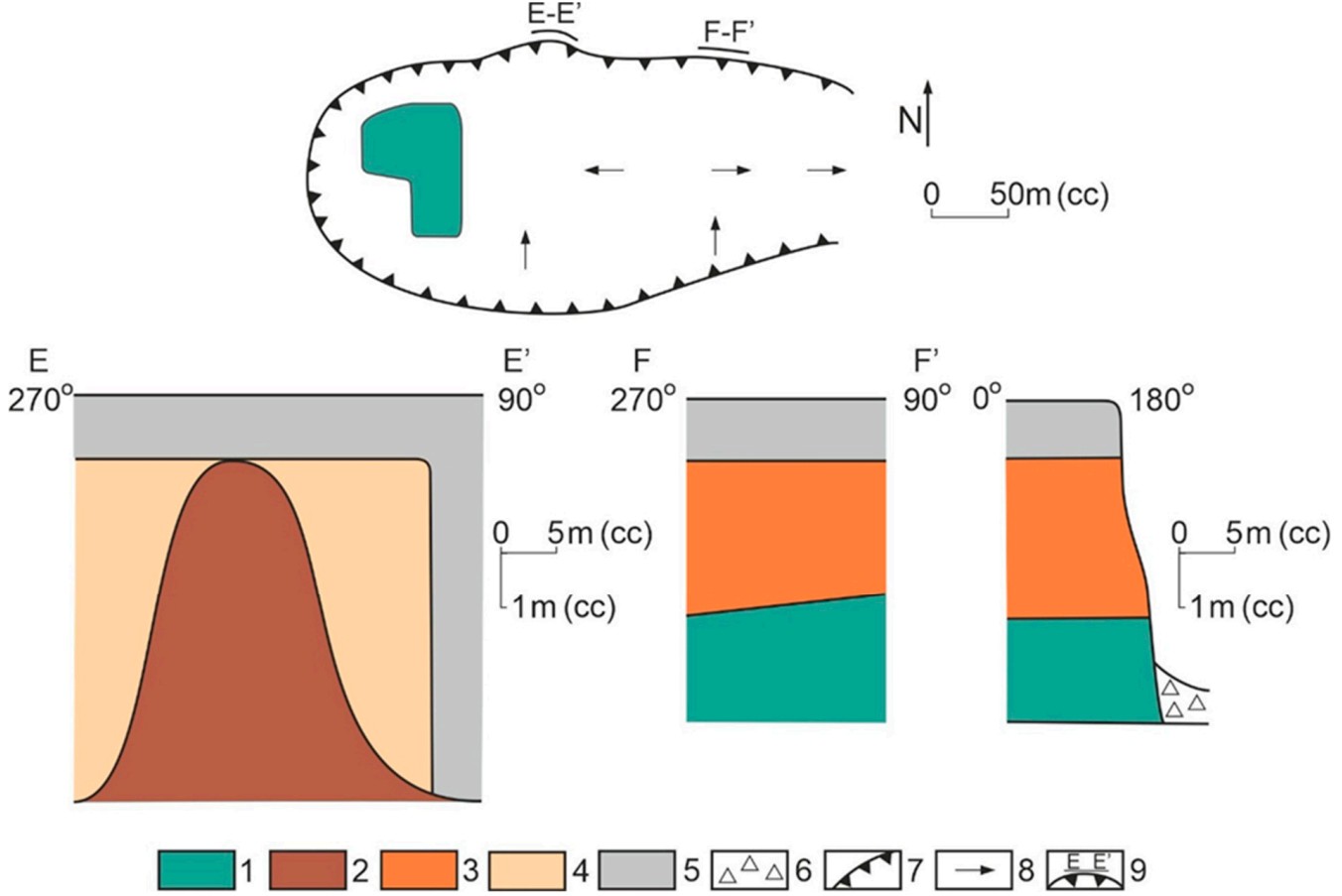

**Figure 9.** Profiles from the northern wall of Márkó mine. Legend: 1. solid dolomite, 2. coarse-grained rubble, 3. rubble of medium grain size, 4. fine-grained rubble, 5. soil and superficial deposit, 6. fallen debris, 7. mine margin, 8. direction of surface inclination, 9. mine margin and profile site.

The rubble mine of Márkó is not a mine, but it is separated into two strip pits (Figure 4). The mines are located between main road 8 and Stream Séd. Mine Márkó 1 was made on the slope of the tributary valley of Stream Séd. Its margin is more elevated than the channel of Stream Séd by about 20 m. The mine is surrounded by superficial deposits (waste dump), clay, gravely clay, and loess that was piled up in entrenchments. The mine is elongated in the EW direction and its floor is separated into several levels (Figure 7A). Its extension is about 500 × 250 m. On the mine walls, there had also been paleokarst depressions with reworked bauxite clay fill that are now denuded. Yellow discolorations also occur in beds of pit-like development, but of horizontal bedding and in patches (Figure 11B). The mine walls terminate in rubble with the exception of some parts of the northern side. Solid rock can be found below the dolomite rubble in the middle of its northern side slope (Figure 9). Commonly, rubble of medium-grain size and fine-grained rubble alternate horizontally. In this case, the rubble boundary is almost vertical and straight, in other cases, the boundary is horizontal but dissected by karst mounds and depressions (Figure 9). Some parts of the mine floor are constituted by solid rock (Figure 9).

The mine of Veszprém is westwards from the town of Veszprém (Figure 4). It is in the WNW-ESE direction, complex, and separated into two strip pits. It has a strip pit in the WNW-ESE direction which is complex, separating into two parts. An inner strip pit was made in the western part of the strip pit. In its eastern part, wetland vegetation can be found. Its extension is about 100 m × 200 m. Today, it is an abandoned mine.

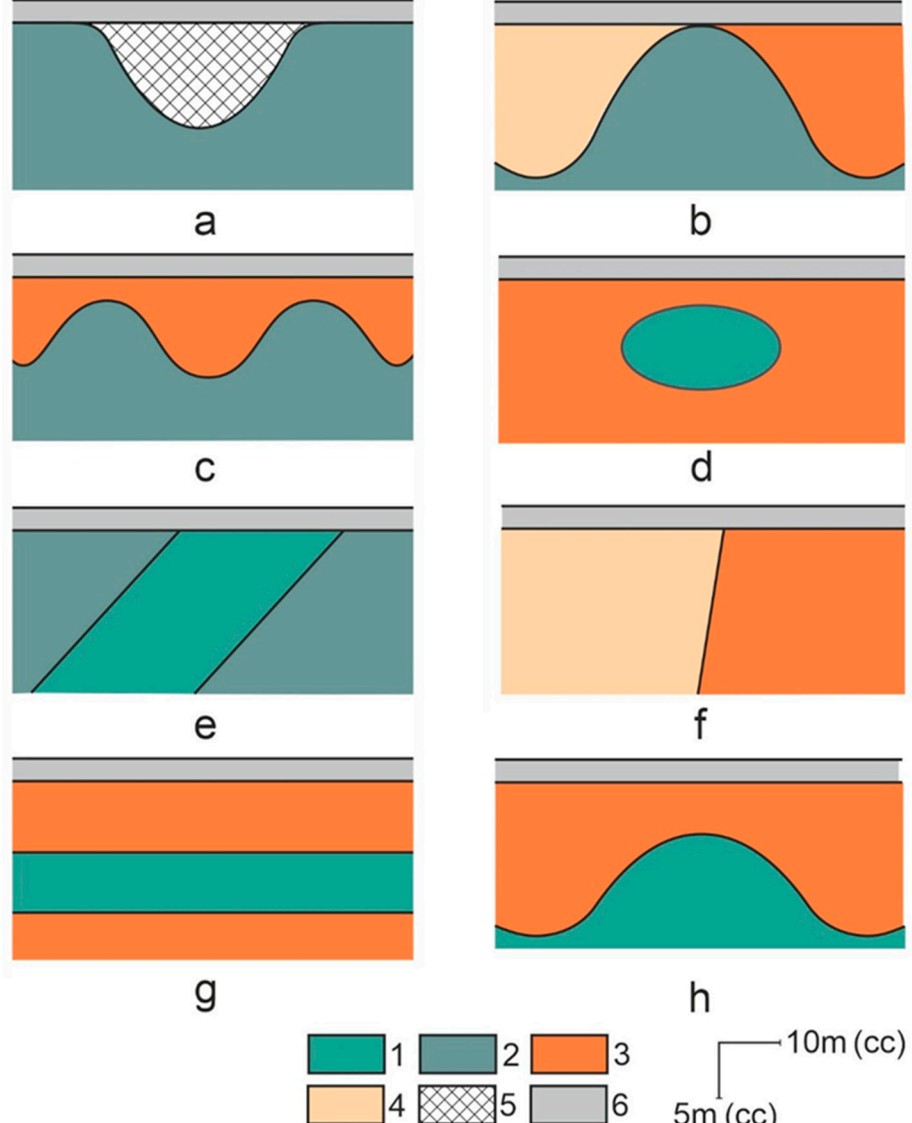

**Figure 10.** Rubble structures in a rubble mine excavated by various mechanical mining methods. Legend: (**a**) depression with bauxite or clay with bauxite fill (Litér mine), (**b**) dolomite, or dolomite rubble mound (Márkó 1 mine), (**c**) dolomite dissected by mounds and depressions or dolomite surface with coarse-grained rubble (Márkó 1 mine), (**d**) solid rock block in rubble (Kádárta mine), (**e**) inclined solid rock enclosed by rubble (Márkó 1 mine), (**f**) horizontal interface of various rubble beds (Márkó 1 mine), (**g**) vertically alternating solid rock and rubble (Márkó 1mine), (**h**) dolomite mound (Sóly mine), 1. solid dolomite, 2. coarse-grained rubble, 3. rubble of medium grain size, 4. fine-grained rubble, 5. bauxite or clay with bauxite, 6. soil and reworked clayey superficial deposit.

The Litér mine (Figure 7B) is in the southern direction from main road 8. Its margin is more elevated than the Séd Stream by about 20–40 m, it is elongated in the EW direction, and its extension is 1000 m × 500 m. The strip pit constitutes several levels. Its margins are straight and a tributary mine forming an arcuate embayment occurs only at one place. The clayey cover has a thickness of several meters. On the north-western mine wall, a depression with reworked bauxite fill occurs. On several sites of the mine walls, rubble beds of yellow discoloration and vertical development occur (Figure 11A).

The Kádárta mine is in the northern direction from main road 8. It is elongated in the ENE-WSW direction and its extension is 300 m × 500 m. The caprock is thin and gravel is absent in its environs. Its margins are straight and a tributary mine forming an embayment only occurs at one place. There is solid rock with block development, where the block with

an extension of 10 m is enclosed by rubble from every direction (Figure 10d). The rubble has yellow discolorations in this mine too.

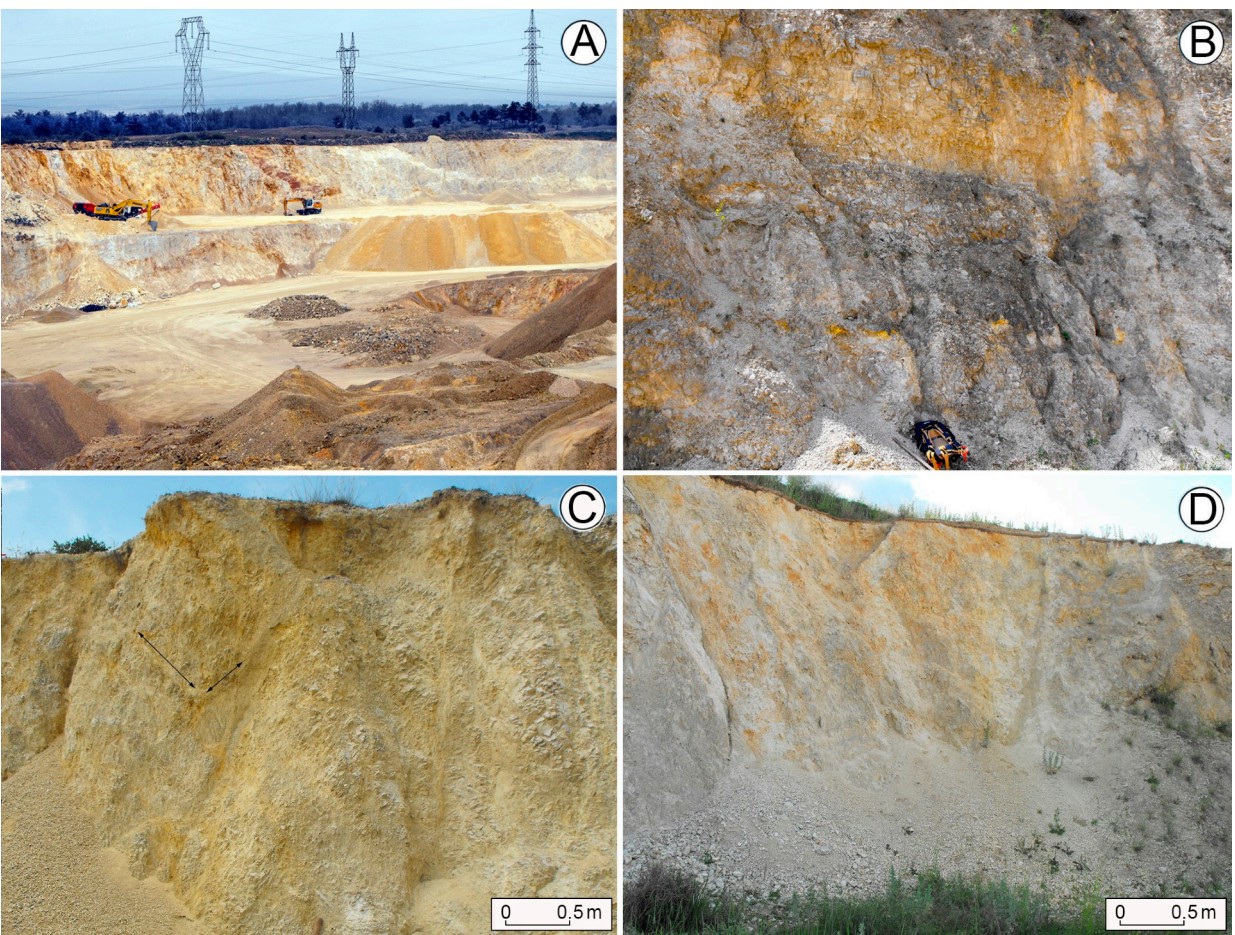

**Figure 11.** Yellow discolorations in the rubble: (**A**) pit-like development (Litér mine), (**B**) horizontal and pocket-like development (Márkó1 mine), (**C**) beds with bending bedding (mine of traditional mining near Sóly, the arrows show the yellow bending beds), (**D**) patchy development in the fill of the karstic depression of Kádárta mine.

The Hajmáskér mine is separated into two strip pits, both of which are in the EW direction. The western one has a narrow ground plan, while the eastern one has an almost round ground plan. It is an abandoned mine nowadays.

The Sóly mine is between main road 8 and the Szombathely-Budapest railway line. Its margin has a 20 m elevation relative to Stream Öskü. The ground plan of the mine is slightly elongated in the EW direction. In its environs, the caprock is not thick (only 1–2 m) but contains a lot of gravel. Its strip pit is one-level and plain. In the stripe pit that was created in rubble, the island-like patches of the solid rock constitute mounds (Figure 8).

The material of dolomite (rubble) is discontinuous non-solid rock, which occurs at different altitudes and morphological environments and has the following varieties in the studied area.

The dolomite is fractured when the rock is very thin and of irregular pattern with coalescing fractures. It cannot be disintegrated into parts with tools either. This type is regarded as pre-rubbled dolomite. Coarse-grained rubble can only be disintegrated into parts with tools (hammer or pickaxe) and with hits. The size of the separated parts can reach 1–2 dm. The parts are elongated in one direction. Medium-grained rubble can be separated by hand and is constituted by parts with a maximum size of approximately

4–5 cm. The parts are angulate, and their size is similar in every direction. Parts of the fine-grained rubble have a diameter smaller than several millimetres.

Other rubble varieties are mixed rubble and soil with rubble. In the case of mixed rubble, rubble of medium grain size and fine-grained rubble or coarse-grained rubble and rubble of medium grain size occur together in a mixed way. In the case of soil with rubble, in the soil, there are grains that are equal to the rubble of medium grain size. These beds might have developed in a way where the rubble became soil or debris of frost weathering may have got into the soil.

In various exploitations, yellow discolorations can be distinguished too. The yellow material is fine-grained clay, it contains fine-grained rubble in different proportions. Its bedding can be vertical pit-like (Figure 11A), horizontal (Figure 11B), separated into pockets (Figure 11B), and bed-like bending in an arcuate way (Figure 11C).

In the studied area, rubble occurs on two levels: in the upper level and in the lower level. The rubble of the upper level is mainly from traditional mining, while that of the lower level is mostly exposed in mines excavated by mechanical mining. The rubble of the upper level is mostly at a more elevated position than the rubble of the lower level, and its thickness is thin at several meters. At its occurrence, soil or soil with rubble occurs at the surface. The rubble of the lower level is at a less elevated position than that of the upper level. Based on the data of the mines excavated by mechanical mining, the rubble of the lower level is thicker (several meters). At its occurrence, the surface with rubble is covered with clay or clayey caprock.

Beds that underwent rubble formation in various ways can be well distinguished. In the mines, rubble beds and solid dolomite have some beds that are specific regarding rubble formation, these are the following.

- Vertically, the grain size of rubble beds is of two types: it can become finer upwards (normal bedding) or it can become coarser upwards (inverse bedding). In mines of mechanical mining, it is mostly coarser and coarser upwards (Figure 12). In the walls of mines of traditional mining, the rubble becomes finer upwards (Figures 13 and 14).
- Non-mixed rubble beds can be well distinguished from each other.
- Solid rocks give the extension boundaries of the mines in some of their parts, for example in the case of the Sóly mine (Figures 8B and 9). In other sites, the mine does not terminate at the solid rock, but the margin of the mine is constituted by rubble.
- It occurs that the solid rock gives the floor of the strip pit or at least a part of it (Figure 9). It may also give the margin of the mine, but not at the upper margin of the slope, but at the middle or lower part of the slope, and the rubble is situated above this level (Figure 9).
- The solid rock can also be wrapped. In this case, blocks of various sizes are enclosed by rubble beds in the excavations (Figure 10).
- Dolomite may also be of island-like development. In this case, rubble encloses dolomite patches which constitute mounds in strip pits (Figure 8A).
- The rubble beds may have preserved the morphology of the former karstic surface (Figure 10). In this case, reworked bauxite, clay with bauxite, clay, gravely clay, and loess was transported into the indentations of the rubble surface by fluvial transportation, mass movement, and pluvial reworking.
- Different rubble varieties may be distinguished by karstic surfaces.
- Different rubble beds may alternate not only vertically, but also horizontally (Figure 10f). The transition between different rubble beds that are contiguous with each other may be gradual or sharp. The surface of the interface is subvertical.
- In the mine walls, there may occur beds at the same elevation and at the same level which underwent rubble formation to various degrees (the rubble is of different grain sizes).

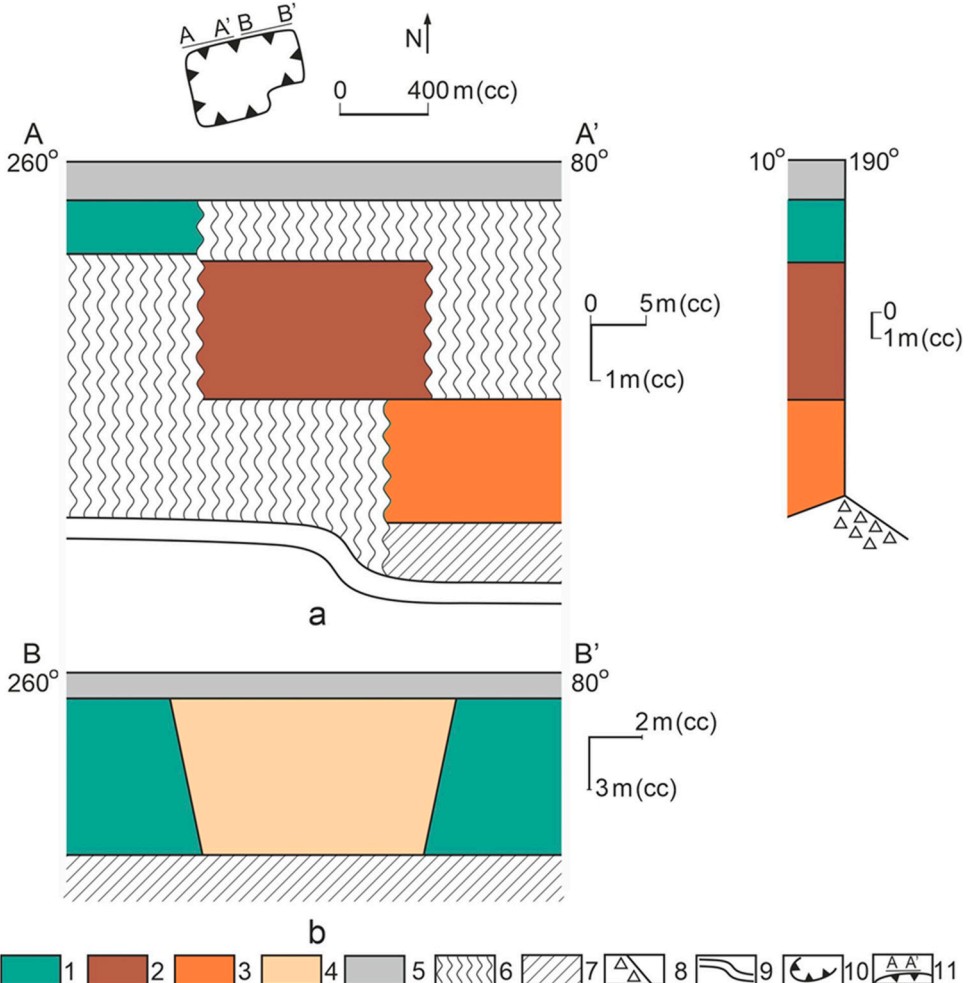

**Figure 12.** Inverse dolomite rubble beds (**a**) which is at the site of A-A' profile and the fill of a filled karstic depression (**b**) which is at the site of B-B' profile of Kádárta mine. Legend: 1. dolomite, 2. coarse-grained rubble, 3. rubble of medium grain size, 4 fine-grained rubble, 5. soil and superficial deposit, 6. dump pushed onto the mine wall, 7. slope debris in front view, 8. slope debris in lateral view, 9. mine road, 10. mine, 11. profile site.

When comparing rubble and non-rubble dolomite samples (Table 3), it can be established that in spite of the fact that the sample sites can be regarded as accidental, the proportion of calcareous dolomites is higher even relative to all samples or to samples of solid dolomites than in the case of those that originate from rubble (Tables 3 and 4). It can also be stated that while no dolomite of compact texture occurs in the samples of rubble beds, non-rubble beds contain dolomites with compact texture (Table 4). Regarding samples of compact texture, it can be established that on non-rubble dolomites, either in the case of calcareous or non-calcareous samples, dolomites of compact texture occur in significant proportions. Their proportion is 27% relative to the proportion of all samples, while it is 28.29% relative to samples originating from non-rubble dolomites.

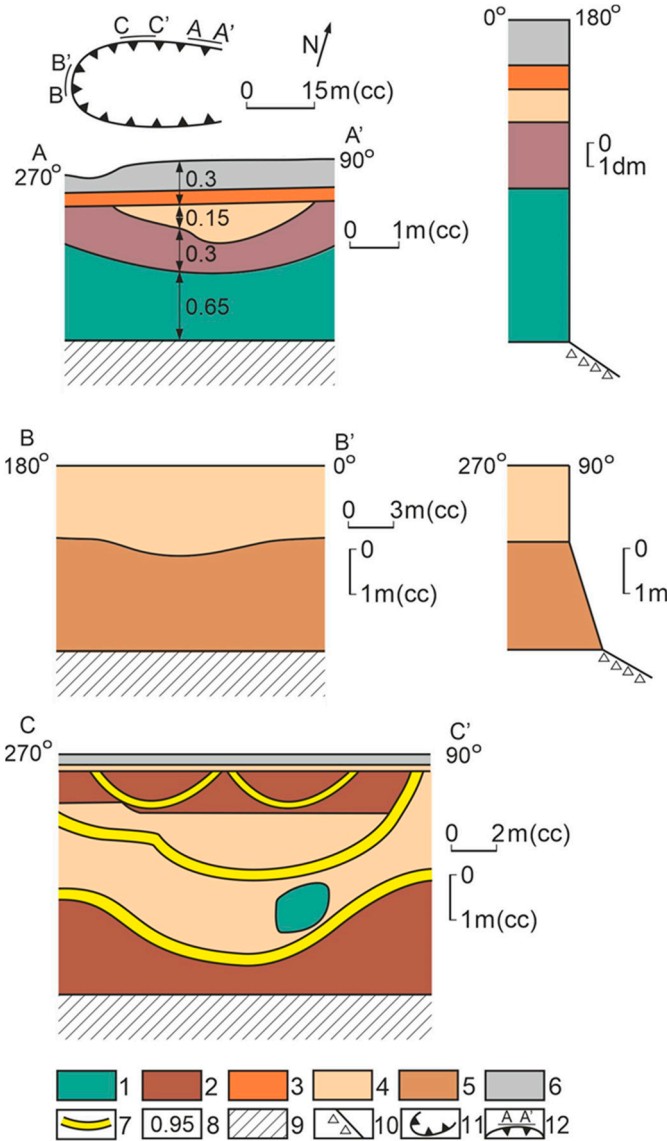

**Figure 13.** Profiles of a mine excavated by traditional mining: The sites of profiles can be seen in overview map. Legend: 1. dolomite (solid rock), 2. coarse-grained rubble, 3. rubble of medium grain size, 4. fine-grained rubble, 5. mixture of coarse-grained rubble and rubble of medium grain size, 6. soil, 7. yellow discoloration in the rubble, 8. thickness data in meters on the A-A' profile, 9. debris slope of the wall in front view, 10. debris slope of the wall in lateral view, 11. mine, 12. profile site.

Mine margins above the channel of Stream Séd are situated at a low relative height (Table 5). Therefore, height differences are between 10 m and 40 m. The floors of strip pits are situated above Stream Séd with only some meters. The majority of mines currently operating have not reached the karstwater level since, with the exception of the Veszprém mine, there is no permanent water on their floors. However, lakes occur (for example Kádárta Fishing Lake), which refers to the fact that the mine situated here had to be abandoned due to the appearance of karstwater. Therefore, strip pits are close to the present karstwater level.

**Table 3.** Calcareous dolomites, non-calcareous dolomites, dolomites with rubble, and dolomites without rubble based on 124 samples in the Bakony Region (based on the sample data of Hegyiné Pakó et al.) [2].

| | |
|---|---|
| **without rubble, calcareous** | 8(mg,t), 9(mg), 26(f), 30, 32(mg,f), 34(t,k), 35(t), 36, 38, 39(k), 40(k), 41, 48, 49(t), 55, 57(f), 58(f), 59, 63(t), 89(k), 90(t,k) |
| **all calcareous** | 21 (out of them 5 is compact) |
| **without rubble, non-calcareous** | 1(b), 2(b), 3(k), 4(k,t), 5(kt), 6, 7(b), 12, 16, 17(k), 18(t,k), 20(k). 21(t), 22, 23, 24(tr), 25(f), 27(f), 28(t,f), 29(t,f), 31(f), 33(t), 37(t), 43, 44(t), 46(f), 47(f), 50, 51(t), 52(t,f),53(f), 54(t,f), 56(f), 66(k), 74, 75, 76, 77, 78(k), 79(k), 80(k), 82(t), 83(t), 85(t), 86, 87(k), 88(k), 91(k), 92(t,f), 93(f), 94(t), 95(f), 96(t,k), 103(k), 106(k), 107(t,k), 109(t), 110(t), 113, 114, 115, 117, 118(k), 119(k), 120(k), 121(k), 122(k), 123(k), 124(t,k) |
| **all (calcareous and non-calcareous)** | 90 samples of non-rubble dolomite, 26 samples of non-rubble with compact texture<br>69 samples of non-calcareous, non-rubble (out of them 21 samples are compact) |
| **with rubble, calcareous** | 45(tr), 61(tr), 68(tr,k), 69(b,h,k), 81(b,h,k) |
| **all with rubble and calcareous** | 5 samples |
| **with rubble, non-calcareous** | 10(b,p), 11(tr,h,k), 13(p), 14(b,h), 15(b,h), 19(b,h,k), 42(b,h), 60(tr,k), 62(p), 64(m,k), 65(tr,k), 67(tr), 70(b,h,k), 71(tr,k), 72(b,h,k), 73(b,h,k), 84(b,h,k), 97(tr,k), 98(p,k), 99(tr,k), 100(b,h,k), 101(b,h), 102(m), 104(tr,k), 105(tr,k), 108(b,h), 111b,h), 112(p,h,k), 116(p) |
| **all with rubble, non-calcareous**<br>**all with rubble (calcareous and non-calcareous)**<br>**all calcareous with rubble** | 29 samples are with rubble, 0 samples compact in the rubble<br>34 samples<br>5 samples |

h: assumed as being of warm water origin; f: drilling; b: non-solid; p: powder-like; m: with rubble; tr: fractured; t: of compact texture; mg: marly; k: mine; the numbers are the sample numbers used by the authors [2].

**Table 4.** In solid dolomites and dolomites with rubble, the proportion of calcareous dolomites and dolomites of compact texture (%).

| Relative to All Samples (124) | | | | | Relative to All Solid Rocks | | Relative to All Dolomites with Rubble | |
|---|---|---|---|---|---|---|---|---|
| Non-Calcareous, Solid | Calcareous, Solid | of Compact Texture | Non Calcareous Dolomite with Rubble | Calcareous Dolomite with Rubble | Non-Calcareous Solid Dolomite | Calcareous, Solid Dolomite | Non-Calcareous Rubble Dolomite | Calcareous Dolomite with Rubble |
| 55.65 | 16.94 | 28.89 [1]<br>20.97 [2] | 23.39 | 4.03 | 55.65 | 16.94 | 85.29 | 14.71 |

[1] dolomite of compact texture relative to all solid dolomites; [2] dolomite of compact texture relative to all samples.

The fracture density of non-rubble dolomites with valley side positions is lower (average density is 0.92, the number of sample sites is 12), while in the rubble environment it is higher (average fracture density is 3.34, the number of sample sites is nine). In rubble areas, fractures with the smallest width (their width is below 1 mm) wedge out; therefore, they cannot be regarded as having a tectonic origin.

**Table 5.** In mines excavated by mechanical mining, the altitude of the surface and the altitude of Stream Séd in their environs, and the altitude of the karstwater level.

| Mine | Altitude of Its Margin [m] | Altitude of Stream Séd [m] | Altitude Difference of the Mine and Stream Séd [m] | Altitude of Karstwater Level | Altitude Difference of the Mine and the Water Level [m] | Notice |
|---|---|---|---|---|---|---|
| Márkó1 | 300 | 280 | 20 | 270? | 30 | |
| Márkó2 | 310 | 280 | 30 | 270? | 40 | |
| Veszprém | 250–260 | 240–250 | 10 | 240–250? | 10 | its floor is waterlogged, marshy |
| Kádárta | 220–240 | 180–200 | 40 | 190–200 | 30–40 | |
| Litér | 220 | 180–200 | 20–40 | 170–180? | 30–40 | |
| Hajmáskér | 200–220 | 160–180 | 40 | 170–180 | 30–40 | |
| Sóly | 200 | 180–200 | 20 | 150–160 | 40–50 | Stream Öskü instead of Stream Séd |

altitude of karstwater level, the value preceding karstwater extraction,? = not certain

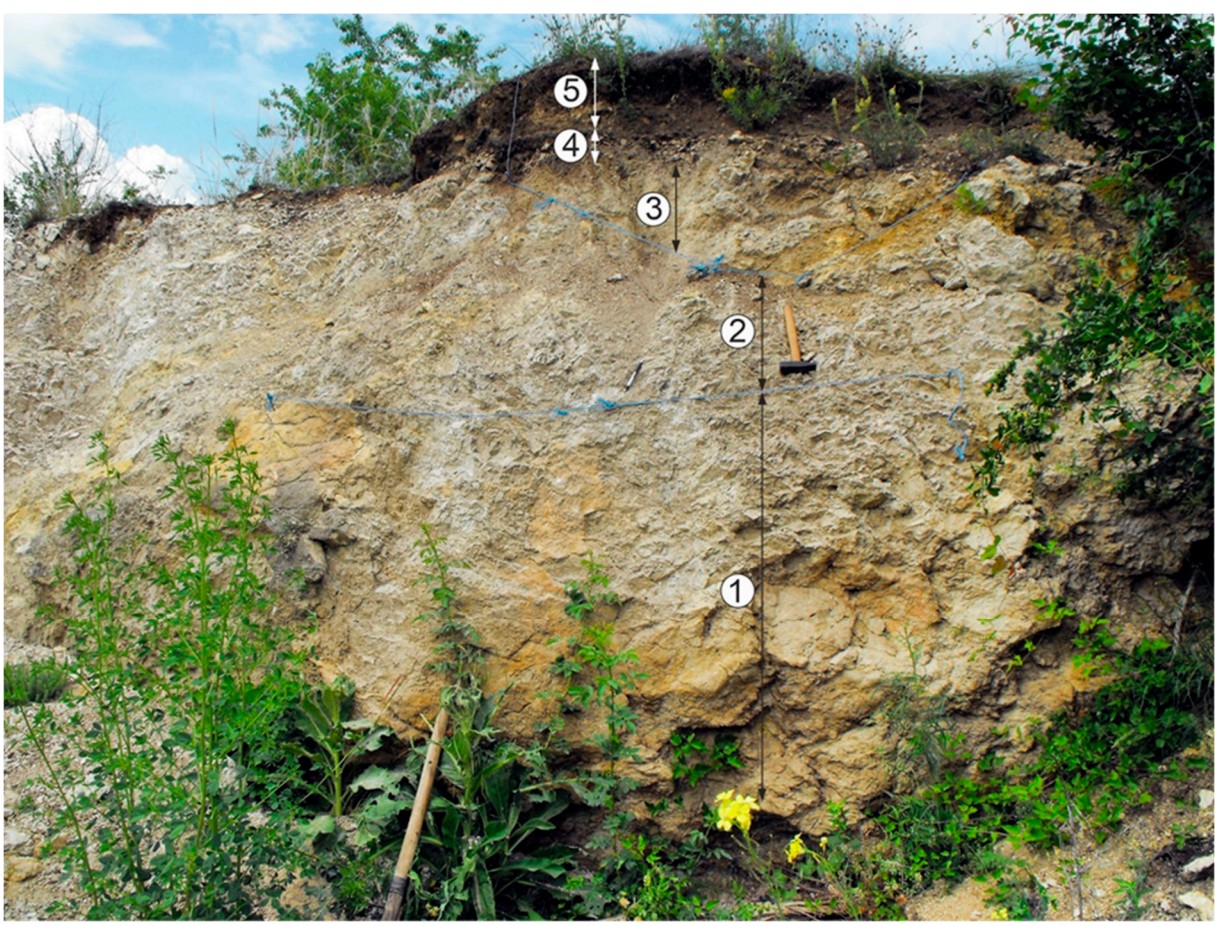

**Figure 14.** Photo of the A-A′ profile of the mine excavated by traditional mining. Legend: 1. dolomite (solid rock), 2. coarse-grained rubble, 3. fine-grained rubble, 4. rubble of medium grain size, 5. soil (with rubble).

## 5. Discussion

According to Jugovics [40], rubble formation takes place due to the effect of microorganisms. Badinszky [41] states that rubble formation is a phenomenon of microtectonics. Pécsi [32] explained the development of the rubble in the studied area by frost weathering. Jakucs [42] states that rubble formation takes place by the dissolution of the rock. During dissolution, the calcareous content of dolomite is first dissolved (which cements the dolomite crystals together), then the dolomite crystal is dissolved (first its Ca ion and finally its Mg ion). Rubble formation takes place through the dissolution of calcareous material according to Waltham et al. [43]. However, rubble formation depends not only on calcareous content, but also on texture. According to Jakucs [42], dolomites of compact texture (almost amorphous substances) do not undergo rubble formation because calcium carbonate and magnesium carbonate are present in uniform dispersibility in the rock and thus, water cannot perform selective dissolution. However, Jakucs states that warm water mechanical weathering can be traced back to different processes (for example volume change), but according to another opinion, the weathering of the dolomite is due to the dissolution effect of warm waters [2]. Veress and Szabó [34] applied the theory of rubble formation of dissolution origin for the studied area.

Although there are signs of frost weathering on outcropping dolomite cliffs, the origin of frost weathering is excluded by the great thickness of the rubble since in mines excavated by mechanical mining, the thickness of rubble may be several tens of meters. In Nyirád (Southern Bakony), which does not belong to the studied area, the thickness of rubble reaches 50 m [14]. However, fine (powder-like) rubble is also hard to be explained by frost weathering. The relatively small thickness of rubble (mines of traditional mining) also excludes the warm water origin at some sites. On the floor of the strip pit, solid rock occurrences are present, above which for example at mine Márkó 1, fine rubble is in the mine wall (Figure 9). At this mine wall, there is no possibility of hot water upwelling due to the presence of solid rock. In their study, Hegyiné Pakó et al. [2] found a relationship between rubble formation, yellow discoloration, and warm-water effect of the samples. However, fine rubble occurs at several sites where no yellow discoloration occurs. Additionally, the yellow discolorations are not always pit-like developments, but they are often bed-like (they may be inclined) or in horizontal extensions, the discoloration of pockets constitutes rows (Figure 11). These facies cannot be interpreted by the former presence of warm water upwellings. The above-mentioned authors mention two samples from the margin of a spring pit Cserszegtomaj (Samples 12, 13, Table 1) which did not discolour. There must have been a higher probability of warm-water effects, but no discoloration took place. However, in the case of another sample, yellow discoloration occured, but there is no rubble formation (Samples 22, 28, Table 1). Finally, in this area of large extension, rubble has a significant distribution and at many places of continuous development. Warm-water effect of such great extension, associated with spring emergence, can be excluded. Badinszky (1973) rejects the idea of the thermal water effect, but he mentions the role of lukewarm water in powder formation. He regards powder formation as a phase of dolomite clay formation. The material of yellow discolorations is clayey.

However, the CaO/MgO proportion is different in the mountains because the proportion in the percentage of MgO mass is different in various samples (Table 1). It is only possible if the magnesium ion has been dissolved to a larger degree than the calcium ion. Since the dissolution of the magnesium ion increases with temperature growth, the dolomite of the whole mountains was affected by a heat impact which enabled the dissolution of the magnesium ion to a larger degree, independently of the fact whether it was rubble or non-rubble. As a result of the large regional extension of high-temperature dissolution, it is not a hot spring, but the dissolution was due to the karstwater. The warming up of the karstwater can be associated with the basalt volcanism of the Balaton Uplands which took place in the Upper Pannonian [29]. Rubble formation took place independently of this effect, probably subsequently.

Among the reasons for rubble formation, area-specific and non-area-specific reasons are distinguished. The former only affect the studied area, the latter may occur anywhere. Area-specific reasons may be fracturing density (however, as mentioned below, it partly originates from rubble formation too), texture structure, and calcareous content. The reason for the great fracture density can partly be explained by the tectonics of the area. In the area, two reverse faults in Litér and Veszprém can be detected [44], which are in the NE-SW strike direction and enclose the area. Therefore, the studied area is a compression belt, where there is a greater chance of the development of fractures that are formed due to pressure than elsewhere.

The characteristics of the rhombohedral crystal of the dolomite contribute to rubble formation, particularly if it is of a non-compact texture. (Crystal characteristics are mentioned among non-specific characteristics because in other dolomite areas of the Bakony Region, dolomites of non-compact texture can also be found.) In the rhombohedral crystals of the dolomite, cation planes constituted by calcium and magnesium ions are aligned [45]. Due to the presence of the planes, the crystal fractures well, but joint surfaces may also develop within the crystal by the dissolution of calcium ion planes.

The frequencies of calcareous and non-calcareous samples support the rubble formation of Jakucs's opinion [42]. Samples with calcareous content are more common in non-rubble dolomites than in dolomites with rubble (Tables 3 and 4) because rubble formation is accompanied by the dissolution and thus, the decrease of calcareous content. The lack of samples of compact texture refers to the fact that at these sites, selective dissolution took place which favoured rubble formation.

Dissolution is proven by the fact that epikarst also occurs in a rubble environment on dolomite, the cavities of which could only have developed by dissolution, but here, the rock did not undergo rubble formation (therefore, mining was stopped at these sites in the mines). However, in their environs there is and was epikarst without rubble. Cavity formation and rubble formation are the results of the same process, dissolution.

Rubble formation is proportional to calcareous content and the quantity of dissolved and transported-away material. In addition to $CO_2$, this is controlled by the duration of dissolution and the intensity of water motion. At a given site, the more intensive and of longer duration the dissolution, and the more intensive the transportation away of dissolved material, the finer-grained the rubble. Dissolution starts along tectonic fractures, which accelerates water motion and material transport. To the effect of faster water motion, dissolution also takes place in the gaps between fracture-free crystals. Newer surfaces develop (these are surfaces intersecting each other) which contribute to faster water flow and further material loss. Along these surfaces and fractures, the rock becomes loose and the parts become separated from each other. Therefore, the fracture density of rubble dolomites and non-rubble dolomites is different (Figure 15). Dolomite rubbles of different grain sizes represent different stages of this process. However, the relationship between rubble formation and dissolution is also proved by the compactness distributions of the samples. In non-rubble samples, the proportion of those with compact texture is high, which hinders the dissolution of calcium carbonate.

Rubble beds with a thickness of several tens of meters cannot be explained by only the dissolution effect of meteoric water, but thin rubble beds that are coarser and coarser downwards can (Figure 16). In addition, the rubble beds of the mines with great thickness are of inverse bedding (Figure 12) thus, they are finer grained at the bottom. If greater dissolution intensity results in smaller rubble size, the dissolution effect must have affected the rock and migrated it upwards, while in the case of normal bedding, it spread from the top downwards (Figures 13 and 14). Although the margins of the mines excavated by mechanical mining are situated above the current karstwater level, the small altitude difference between Stream Séd and the mines, the degree to which some mine floors (or former mines) are covered with water refers to the closeness of the water level. The current level of karstwater might have subsided into its present position as a result of the rise of the area and the deepening of Stream Séd and might have been closer to the surface earlier. The

vertical movements of the karstwater and the fluctuations of the karstwater level resulted in the vertical widening of rubble formation.

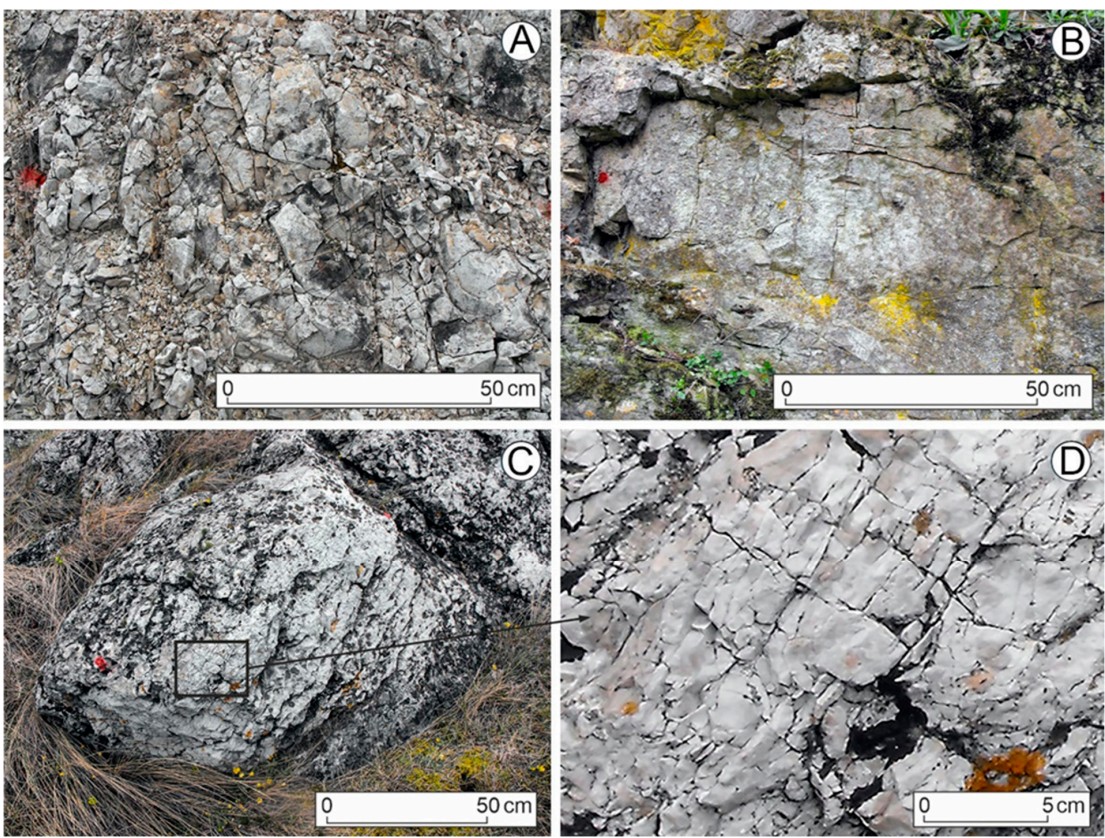

**Figure 15.** Fractures: (**A**): Fractures of medium width in rubble environment (Sóly, road cut), (**B**) fractures of medium width in non-rubble environment (Tekeres valley near Veszprém), (**C**) wide, medium, and narrow fractures in rubble environment (in an abandoned mine of traditional mining near Hajmáskér, Séd), (**D**) an enlarged detail of photo (**C**).

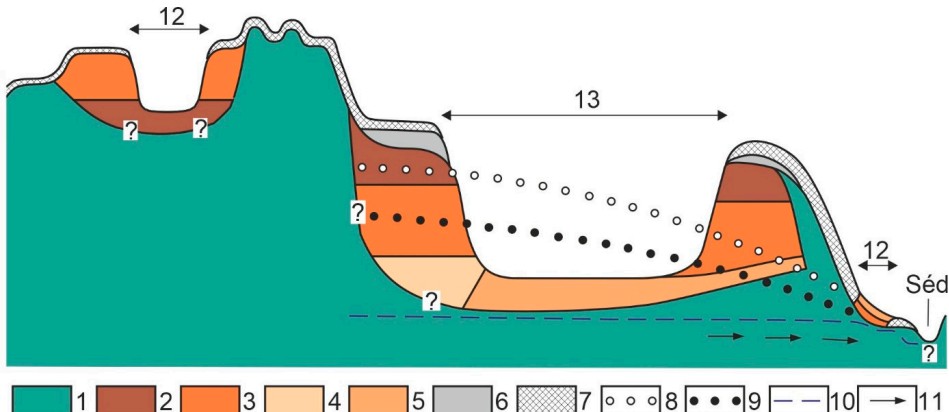

**Figure 16.** Rubble beds of mines of different types. Legend: 1. dolomite (solid rock), 2. coarse-grained rubble, 3. rubble of medium grain size, 4. fine-grained rubble, 5. mixture of rubble of medium grain size and fine-grained rubble, 6. reworked clayey superficial deposit, 7. soil with rubble, 8. former upper karstwater level, 9. former lower karstwater level, 10. present elevated karstwater level, 11. karstwater flow, 12. mine excavated by traditional mining, 13. mine excavated by mechanical mining. ? presumed rock boundary

In the case of dissolution of karstwater origin, dissolution depends on the position of the rock relative to the karstwater level and on the flow rate of the karstwater. At the part of the rock which is closer to the karstwater level and in the case of a great flow rate of long duration, finer-grained rubble is formed since dissolution is of a greater degree. Therefore, the grain size of the rubble may change vertically (but horizontally too) since the karstwater can stay at a certain level for different durations. The altitude distribution of the upper solid rock above the rubble (or coarse rubble) follows the former altitude pattern of the karstwater level (Figure 16). As a result of the different elevations of the karstwater levels, rubble of different grain sizes may be present in the same level or solid rock may even occur at the top.

The lower level, which bears the mines, may subside to a larger and smaller degree. This can be explained by the fact that downwards the pore volume is great in the level of medium-sized rubble, which results in subsequent subsidence (postgenetic subsidence). However, at sites where fine-grained rubble is at the top, surface subsidence can be traced back to the decrease of the rubble, but not to compaction (syngenetic subsidence).

In the case of mines excavated by traditional mining, where the rubble beds are very thin (some meters) and the development of the rubble beds is not inverse, the dissolution effect reached the rock from the surface and spread downwards. Dissolution intensity is the greatest below the soil and thus, the rubble is finer-grained upwards than at a greater depth. Dissolution of meteoric water origin may take place at mines excavated by mechanical mining, but only if the superficial deposit is permeable (loess). However, water can get into the dolomite from impermeable cover too. At such sites, there may be finer-grained rubble at the upper part of the mine. It may occur that the rubble that developed by meteoric water and dissolution of karstwater origin may coalesce. Dissolution of karstwater origin may also be present in the case of mines of traditional mining, at sites where the karstwater moving horizontally crops out (Figure 16). At these sites, rubble development is local, thin, and wedging out. Its development took place downwards (grain size increases downwards) and during vertical water motion (grain size does not change), but waters originating from the surface may also have played a role. At mines excavated by traditional mining, independently of karstwater, dissolution may take place in the case of horizontal flows that developed locally. At sites where the rock undergoes rubble formation since the vertical infiltration of meteoric water slows down in the solid rock, the water is swollen back in the rubble and flows horizontally. Therefore, in mines of traditional mining, the degree of rubble formation may increase and if the flowing water reaches a greater depth, it may happen that finer-grained rubble is formed below coarser-grained rubble. Such rubble formation can be detected in a mine of traditional mining westwards from the settlement of Sóly (Figure 13).

## 6. Conclusions

In the terrain between Márkó and Pétfürdő (Várpalota), the epikarst is of sporadic development in the main dolomite. Instead of the epikarst, the rock underwent extensive rubble formation, the thickness of which is alternating, but it may also be several tens of meters. The reason for rubble formation may be area-specific and non-area-specific. An area-specific reason is high fracture density (but rubble formation also contributes), high calcareous content, and non-compacted texture. The closeness of the karstwater level to the surface and the presence of Stream Séd also play a role, while a non-area-specific reason is crystal structure. As a result of rubble formation, epikarst is missing on the rock. Karren is also mostly absent, which favours mining. The lack of epikarst and rubble formation results in dolomite in the development of a landscape that is different from limestone karsts. The excavations of the mines enable a better understanding of the process of dissolution in the dolomite.

The widespread distribution of rubble is indicated by mines of traditional mining and mines excavated by mechanical mining. Scars being excavated by mines refer to different dissolutions. In mines of traditional mining, dolomite rubble is coarser and

coarser downwards due to the dissolution of surface (soil) origin. In mines excavated by mechanical mining, the dolomite rubble of different grain size has a diverse bedding, but it is coarser and coarser-grained upwards in general. This and the great thickness of rubble refer to dissolution by karstwater. However, dissolution of surface origin may also occur in their area. Therefore, mines of traditional mining may occur at parts of the area of any elevation, while mines excavated by mechanical mining occur on surfaces of lower elevation. At sites where the horizontally moving karstwater appears at the surface (on the floor of the valley of Stream Séd), dissolution of karstwater origin also plays a role in mines of traditional mining, but despite this, the rubble is thin, wedging out, and mixed.

The grain size of rubble depends on the intensity of dissolution and transportation away. Therefore, in the case of dissolution by karstwater, the rubble along the karstwater level is finer. The distribution of fine-grained rubble close to the surface refers to the former more elevated position of karstwater level (paleokarstwater level), or the fluctuation of water level. The excavations of the mines enable a better understanding of the process of dissolution in the dolomite.

**Funding:** This research received no external funding.

**Data Availability Statement:** Data is contained within the article.

**Conflicts of Interest:** The authors declare no conflict of interest.

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
