# Peer review of "Rubble Mines in the Environs of Veszprém (Bakony Region, Hungary)"

_mining, doi:10.3390/mining3040032_

Round 1

Reviewer 1 Report (Previous Reviewer 1)

The article shows the author's significant contribution to its improvement. I therefore recommend the article for publication.

Reviewer 2 Report (Previous Reviewer 2)

The manuscript can be accepted in its present form. 

Reviewer 3 Report (Previous Reviewer 3)

The Author made almost all corrections. All figures and tables describing rubble types in the research area are now in the right positions. The manuscript is now more clear. However the Author did not take into account some comments. But the Author explains in response 3, why these comments were not take into account. The Reviewer accepts the Author's explanations presented in response 3.

In its current form, the article is suitable for publication.

This manuscript is a resubmission of an earlier submission. The following is a list of the peer review reports and author responses from that submission.

Round 1

Reviewer 1 Report

The reviewer thanks the author and editors for the opportunity to review the manuscript.
The manuscript discusses rubble mines near Veszprém from the Bakony region of Hungary. The manuscript has great publication potential, but currently lacks scientific value. The description and characterization of the Rubble mines in the environs of 
Veszprém could be a start for further analysis. Where is the scientific element in this article?

General comments:
1. The introduction and abstract lack information on what the purpose of this article is. The author does not indicate a gap in the research area and it is not clear what purpose the analyses described in this manuscript are intended to serve.
2. The manuscript should be rewritten and prepared according to the standards of scientific articles. Currently, there is chaos in the manuscript, and some tables and figures are not discussed.
3. The author has incorrectly numbered tables throughout the manuscript

Detailed comments:
Abstract and Section 1: The introduction and abstract lack information on what the purpose of this article is. The author does not indicate a gap in the research area and it is unclear what purpose the analyses described in this manuscript are intended to serve.
Lines 24-25: This sentence should be moved to Section 2.
Last paragraph of section 1: On what basis does the author claim that waste dumps are positive features? In general, mining is understood as the destruction of the environment. So the change created by mining causes negative effects in the environment.
Section 3: Remove all hyphens. No description of the tables and figures presented.
Section 4: Rewrite. The author should draw more conclusions from the analyses.

Sometimes the author uses difficult English language that the reviewer does not understand. The English language should be corrected. I recommend using the help of a native speaker.

Author Response

Answer to Reviewer 1

Thank you for your work. My answers are:

-          I have corrected both the abstract and the introduction with new parts in which I emphasized the gap information in literature.

-          I gave more details on the references to figures and tables.

-          I do not know what you mean by the incorrect numbering of tables.

-          I put the first sentence of introduction into Chapter 2.

-          I defined positive features.

-          I added new sections to Conclusions.

Reviewer 2 Report

The paper entitled “Rubble Mines in The Environs of Veszprém (Bakony Region, Hungary)” presents a report on different aspects of the rubbles mined from the location. Unfortunately, the article does not stand for a research article as there is no novel research as such reported here. The whole article, beginning from the title, reads like a project report. The author does not mention how this study is different from the existing ones and how this is going to advance the knowledge already existing. I regret I must recommend against the publication of it.    

Moderate revision of language and presentation required

Author Response

Answer to Reviewer 2

Thank you for your work. My answers are:

  • I added to sections to Results. I listed them in introduction too. I added new parts to Conclusions thus, results are discussed in a more detailed way. I mention that rubble development by karstwater has not been mentioned in literature yet. This enriches substantially the already existing knowledge on the rubble formation of dolomite. The whole study in new since little and sporadic references can be found in international literature. Therefore it cannot be compared with other results since there are no other results, only in Hungarian literature. These have been mentioned and used and redeveloped (e.g. classification of rubble beds according to grain size, and the description of the relationship between grain size and dissolution).
  • This study is research work and not a project. I do not think that it is a project because through the study of an area it describes a phenomenon which can be the basis of general conclusions.

Reviewer 3 Report

Comments for Authors

The article is very interesting. It includes lots of figures and tables describing rubble types in the research area. However, it needs to be corrected and improved to make it more clear. The comments I presented below.

1.      There are lots of interesting figures and tables in the manuscript. However some of them should be placed in different chapters than they are now. Therefore chapters 2. Research area and 3. Methods are too extensive in comparison with chapter 3. Results. In this chapter only one figure is placed and no tables. However, in the Results chapter, the Author refers to the figures and  tables presented in the previous chapters, describing some data in details.

2.      Figure 1 should be include in chapter 2. Research area, not in the beginning of Introduction. At the beginning of the chapter 1. Introduction, information about the purpose of the article should be placed. Then data on rubble deposits included in this chapter by the Author.

3.      Chapter 2. Research Area- the maps with location of deposits should be here but Table 1, including chemical characteristic features of rubble beds should be placed in Results chapter.

4.      Some data, including figures and tables included in the Methods chapter, should be included in the Results chapter, because these are materials prepared by the Author, especially Tables IV and V and Figures from 4 to 15.

5.      The Conclusion chapter is a little bit general, especially the last sentence ”The excavations of the mines enable a better understanding of the process of dissolution in the dolomite and the former movements of karstwater.” The question is why? Because of different type of rubble in the deposits’ profiles, which is possible to observe during excavation? It should be completed.

Author Response

Answer to Reviewer 3

Thank you for your work. My answers are:

  • I reordered the figures based on references and not according to methods.
  • I put Figure 1 to Chapter 2.
  • I do not think that it is necessary to make a map on the cover. On the one hand the cover is often local, for example it occurred at the mines, there they were destroyed therefore no reconstruction is possible. On the other hand, the cover has no significance in rubble formation. Thirdly the cover is not present in all mines of traditional mining thus, no mapping is possible. If I made maps, they could not be described in one single figure. This would increase the number of figures.
  • Table I cannot be put into Results since its data are from literary source. The samples are from the whole mountains and not from special areas.
  • I added new parts to Conclusions. The relationship between karstwater level, fine- grained rubble and water flow is written about in a detailed way.
